# Genetic variation in offspring indirectly influences the quality of maternal behaviour in mice

**David George Ashbrook\*, Beatrice Gini, Reinmar Hager\***

Computational and Evolutionary Biology, Faculty of Life Sciences, University of Manchester, Manchester, United Kingdom

**Abstract** Conflict over parental investment between parent and offspring is predicted to lead to selection on genes expressed in offspring for traits influencing maternal investment, and on parentally expressed genes affecting offspring behaviour. However, the specific genetic variants that indirectly modify maternal or offspring behaviour remain largely unknown. Using a cross-fostered population of mice, we map maternal behaviour in genetically uniform mothers as a function of genetic variation in offspring and identify loci on offspring chromosomes 5 and 7 that modify maternal behaviour. Conversely, we found that genetic variation among mothers influences offspring development, independent of offspring genotype. Offspring solicitation and maternal behaviour show signs of coadaptation as they are negatively correlated between mothers and their biological offspring, which may be linked to costs of increased solicitation on growth found in our study. Overall, our results show levels of parental provisioning and offspring solicitation are unique to specific genotypes.

**\*For correspondence:** david.
ashbrook@postgrad.manchester.
ac.uk (DGA); Reinmar.Hager@
manchester.ac.uk (RH)

**Competing interests:** The authors declare that no competing interests exist.

## Introduction

The close interaction between mother and offspring in mammals is fundamental to offspring development and fitness. However, parent and offspring are in conflict over how much parents should invest in their young where offspring typically demand more than is optimal for the parent (*Trivers, 1974*; *Godfray, 1995*), and the existence of this genetic conflict has been demonstrated in empirical research (*Kölliker et al., 2015*). The resulting selection pressures are predicted to lead to the evolution of traits in offspring that influence parental behaviour (and thus investment). Conversely, parental traits should be selected for their effects on offspring traits that influence parental behaviour indirectly (*Kilner and Hinde, 2012*). The correlation between parental and offspring traits has been the focus of coadaptation models where specific combinations of demand and provisioning are selectively favoured (*Wolf and Brodie III, 1998*; *Kölliker et al., 2005*). The fundamental assumption underlying predictions about the evolution of traits involved in parent-offspring interactions is that genetic variation in offspring exists for traits that indirectly influence maternal investment and vice versa. However, it remains to be shown whether specific genetic variants in offspring indirectly influence maternal behaviour. In an experimental mouse population, we demonstrate that genes expressed in offspring modify the quality of maternal behaviour and thus affect, indirectly, offspring fitness.

To investigate the genetics of parent-offspring interactions we conducted a cross-fostering experiment between genetically variable and genetically uniform mice, using the largest genetic reference panel in mammals, the BXD mouse population. We generated families of genetically variable mothers and genetically uniform offspring by cross-fostering C57BL/6J (B6) litters, in which no genetic variation occurs between animals of this strain, to mothers of a given BXD strain. Conversely, a BXD

**eLife digest** Genes encode instructions that can influence the behaviour and physical traits of the individual that carries them. Individuals of the same species can carry different versions (or variants) of the same gene, leading to a variety of traits in the population. However, a gene expressed in one individual can also alter the traits of another individual. This is known as an indirect genetic effect. For example, a gene in a mother that affects her ability to provide care may influence how her offspring develop.

Researchers have predicted that offspring should be able to manipulate their mothers to try and gain more care than the mothers are willing to give. Furthermore, the offspring born to mothers who respond to this begging are predicted to save energy and beg less. However, few gene variants that indirectly modify the behaviour of mothers or offspring have so far been identified.

Ashbrook et al. used mice to test the idea that genetic variation in particular locations in the offspring's genome can affect maternal behaviour, and vice versa. In the first experiment, mother mice with different gene variants fostered litters of mouse pups that were all genetically identical. In the other experiment, genetically identical mothers fostered litters of mouse pups with different gene variants. Ashbrook et al. identified several locations in the offspring's genome that modified the behaviour of their foster mothers. Furthermore, the experiments also show that genetic variation among the mothers influenced the development of their offspring, independent of the genes carried by the offspring.

The next steps are to identify the specific genes underlying the changes in behaviour, and the molecular and genetic pathways by which they impact indirectly on the traits of other individuals.

female's litter was cross-fostered to B6 mothers (*Figure 1*). Thus, we can analyse the effects of genetic variation in mothers or offspring while controlling for genetic variation in the other. This cross-fostering design has been successfully utilized in previous studies on family interactions because it breaks the correlation between maternal and offspring traits. Here, different families, or naturally occurring variation of maternal and offspring trait combinations across different broods, are assumed to represent distinct evolved strategies (*Agrawal et al., 2001*; *Hager and Johnstone, 2003*; *Meunier and Kölliker, 2012*). From birth until weaning at 3 weeks of age we recorded offspring and maternal body weights and behaviour, following *Hager and Johnstone (2003)*.

## Results

We first investigated whether there is evidence for indirect genetic effects in offspring influencing maternal behaviour, keeping maternal genotype constant. To find out if maternal behaviour is modified by genes expressed in offspring, we mapped variation in maternal behaviour as a function of their adoptive BXD offspring genotype. We found that variation in offspring genotype affects maternal behaviour, which in turn influences offspring development and fitness. Throughout, we denote loci as either maternal or offspring, *Mat* or *Osp*, followed by whether it is an indirect effect (*Ige*) or a direct effect locus (*Dge*). We provide a summary of all loci in *Table 1*, and mapping details in *Supplementary files 1A-D*. During the first postnatal week we mapped a locus on offspring chromosome 7, *OspIge7.1*, modifying maternal nestbuidling on day 6 (*Figure 2*). At this locus, the D2 allele increases the trait value such that B6 mothers showed more nestbuilding activity when fostering BXD pups carrying the D2 allele at the locus. Nestbuilding is particularly important for offspring fitness as thermoregulation is underdeveloped and hypothermia is the primary cause of early death, even if milk is supplied (*Lynch and Possidente, 1978*). We detected a further locus on offspring distal chromosome 5 (*OspIge5.1*) that affects maternal behaviour on day 14, around the time we expect the weaning conflict to be highest (*Figure 3*). Here, mothers showed increased levels of maternal behaviour when fostering BXD offspring carrying the D2 allele. Conversely, we can look at how variation in maternal genotype influences offspring traits, keeping offspring genotype constant, mapping variation in offspring traits as a function of BXD genotype. Here, we found that offspring growth during the second week is affected by a locus on maternal chromosome 17, *MatIge17.1*, where the B6 allele increases the trait value (*Figure 4*). In addition to looking at indirect genetic effects we can

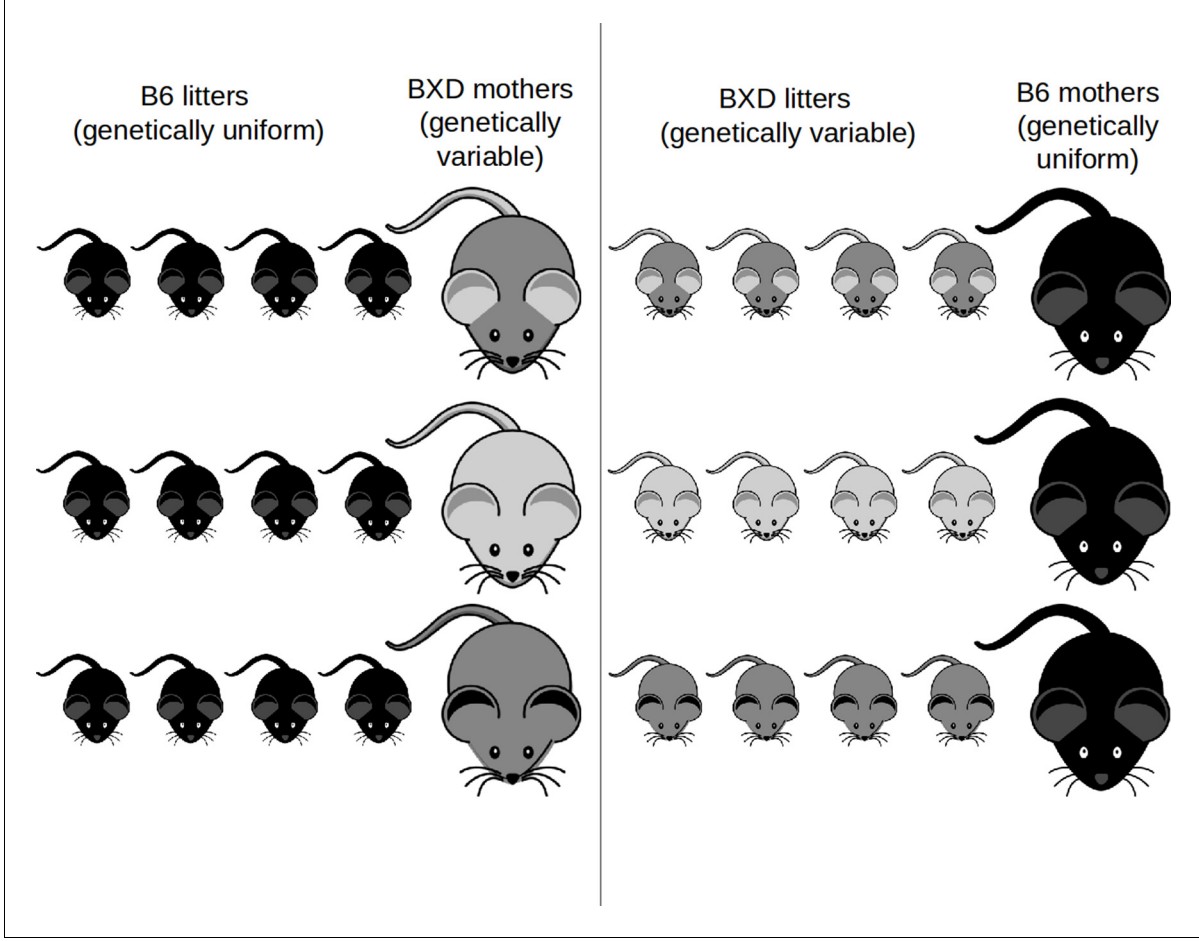

**Figure 1.** Experimental cross-foster design. Females of different lines of the BXD strain (light to grey mice) adopt B6 offspring (dark) and B6 females (dark) adopt offspring born to females of different BXD lines. A total of 42 BXD lines with three within-line repeats plus the corresponding B6 families were set up for the experiment.

analyse direct genetic effects, i.e. how an individual's genotype influences its own traits. In mothers we found a direct genetic effect locus for maternal behaviour on proximal chromosome 10, and for nestbuilding behaviour on chromosome 1 (*MatDge10.1* and *MatDge1.1*, respectively), where the B6

**Table 1.** Summary of direct and indirect genetic effects. We list the loci, followed by their position in Mb, the 1.5 LOD confidence interval (***Dupuis and Siegmund, 1999***), the genome-wide peak marker LRS and LOD score and associated p-value and the number of genes within the interval.

| Loci | QTL position | Confidence interval | Max LRS | Max LOD | Max P | Number of genes |
|---|---|---|---|---|---|---|
| *MatDge1.1* | 168.32 | 165.24 - 172.06 | 17.03 | 3.70 | 0.069 | 68 |
| *OspDge5.1* | 23.827 | 17.82 - 24.62 | 17.85 | 3.88 | 0.046 | 73 |
| *OspIge5.1* | 146.68 | 145.20 - 147.65 | 18.65 | 4.05 | 0.038 | 40 |
| *OspIge7.1* | 53.68 | 47.76 - 56.65 | 17.85 | 3.88 | 0.039 | 232 |
| | 81.49 | 76.12 - 90.92 | 16.06 | 3.49 | 0.087 | 133 |
| *MatDge10.1* | 19.09 | 18.61 - 21.83 | 22.37 | 4.86 | 0.008 | 30 |
| *MatIge17.1* | 23.32 | 11.48 - 31.17 | 19.02 | 4.13 | 0.022 | 422 |
| | 33.02 | 31.32 - 40.65 | 18.57 | 4.03 | 0.028 | 321 |

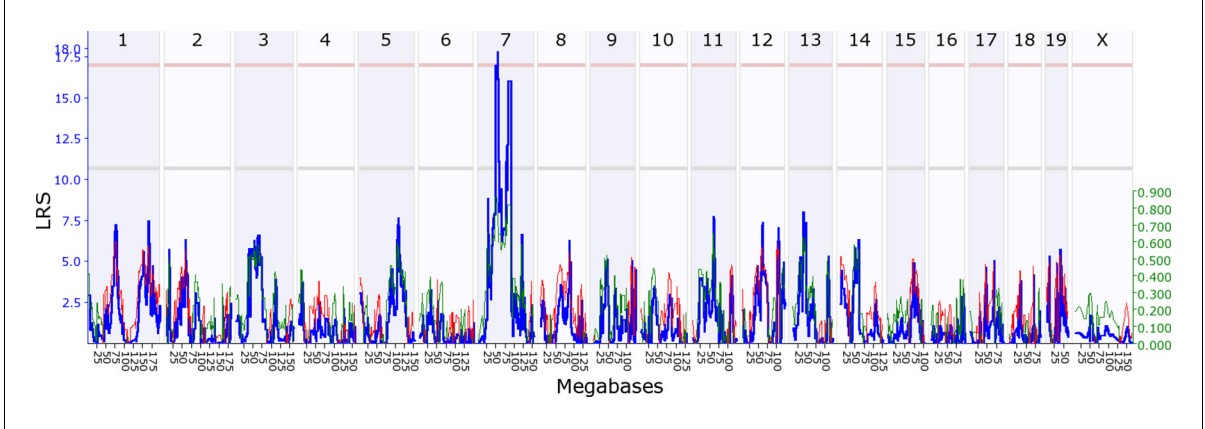

**Figure 2.** Offspring indirect genetic effect modifying maternal nestbuidling behaviour (*OspIge7.1*). The figure shows an offspring genomescan of maternal nestbuilding behaviour on day 6. The blue line represents the genome scan, showing the likelihood ratio statistic (LRS) associated with each marker across the 19 autosomal and the X chromosome. The top, pink, line marks genome-wide significance, the lower, grey, line the suggestive significance threshold. The green or red line show the additive coefficient, with green showing that the DBA/2J alleles increase trait values and red that the C57BL/6J alleles increase trait values. The green axis on the right shows by how much the respective alleles increase trait values (the DBA/2J allele in offspring increases maternal nestbuilding by ~0.8).

The following source data is available for figure 2:

**Source data 1.** Source data for *Figure 2* as uploaded to GeneNetwork, showing the B6 maternal nestbuilding behaviour on day 6 per pup for the BXD lines.

allele increases the trait value in both cases. We also detected a locus for offspring solicitation behaviour on chromosome 5 (*OspDge5.1*) with the D2 allele increasing the level of solicitation shown. Its location is at the opposite end on chromosome 5 from where *OspIge5.1* is located.

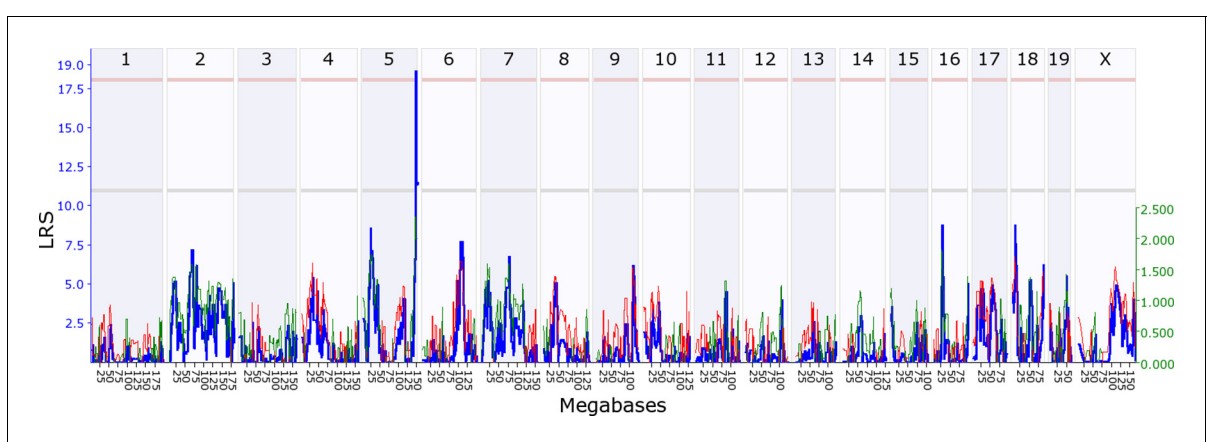

**Figure 3.** Offspring indirect genetic effect modifying maternal behaviour (*OspIge5.1*). The figure shows an offspring genomescan of maternal behaviour on day 14. The blue line represents the genome scan, showing the likelihood ratio statistic (LRS) associated with each marker across the 19 autosomal and the X chromosome. The top, pink, line marks genome-wide significance, the lower, grey, line the suggestive significance threshold. The green or red line show the additive coefficient, with green showing that the DBA/2J alleles increase trait values and red that the C57BL/6J alleles increase trait values. The green axis on the right shows by how much the respective alleles increase trait values (the DBA/2J allele in offspring increases maternal behaviour by ~2.5).

The following source data is available for figure 3:

**Source data 1.** Source data for *Figure 3* as uploaded to GeneNetwork, showing the B6 maternal behaviour on day 14 per pup for the BXD lines.

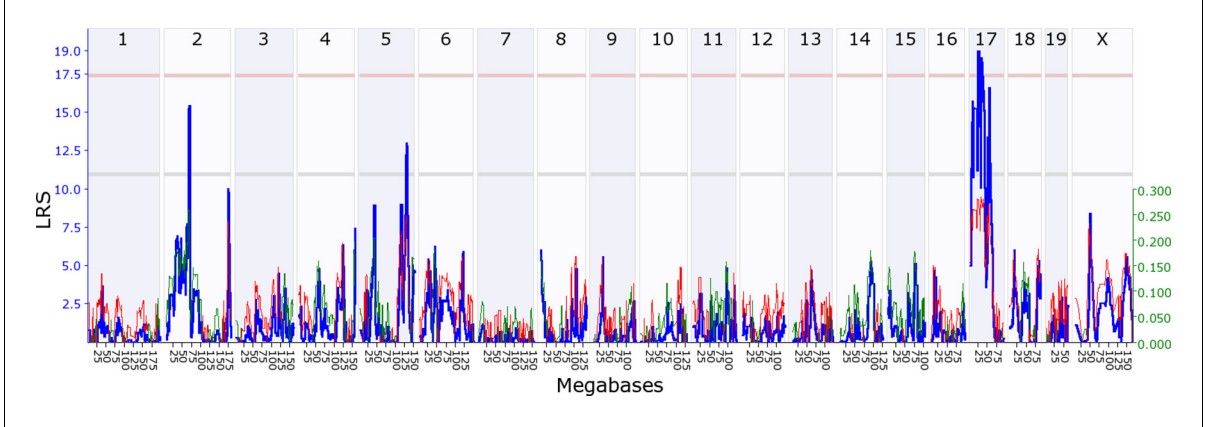

**Figure 4.** Maternal indirect genetic effect modifying offspring growth (*MatIge17*). The figure shows a maternal genomescan of offspring growth in the second postnatal week. The blue line represents the genome scan, showing the likelihood ratio statistic (LRS) associated with each marker across the 19 autosomal and the X chromosome. The top, pink, line marks genome-wide significance, the lower, grey, line the suggestive significance threshold. The green or red line show the additive coefficient, with green showing that the DBA/2J alleles increase trait values and red that the C57BL/6J alleles increase trait values. The green axis on the right shows by how much the respective alleles increase trait values (the C57BL/6J allele in mothers increases offspring growth by ~0.25).

The following source data is available for figure 4:

**Source data 1.** Source data for *Figure 4* as uploaded to GeneNetwork, showing the B6 offspring growth in the second postnatal week per pup for the BXD lines.

Overall, our results show that IGEs, as well as DGEs, can be linked to specific loci that affect parent-offspring interaction, and it is thus possible that selection may occur on genes with indirect and direct effects on parental behaviour. Importantly, our result that variation in maternal behaviour is affected by genes expressed in offspring is also clearly borne out at the phenotypic level: offspring solicitation behaviour in genetically variable BXD pups is positively correlated with the level of maternal behaviour in their genetically uniform B6 adoptive mothers on all three days we measured behaviour with significant effects of maternal behaviour and day on offspring solicitation (GLM, $F_{1,103}$ = 17.62, $P < 0.001$ and $F_{2,103}$ = 14.22, $P < 0.001$, respectively; day 6: Pearsons'$r$ = 0.56, $P$ = 0.003; day 10: $r$ = 0.63, $P$ = 0.001, and day 14: $r$ = 0.55, $P$ = 0.008; *Supplementary file 2, a*). Similarly, we can investigate how traits in genetically uniform B6 offspring correlate with maternal traits in their genetically variable BXD mothers (i.e. within adoptive families). We found that maternal behaviour and day have a significant positive effect on offspring solicitation behaviour (GLM, $F_{1,118}$ = 6.33, $P$ = 0.013, and $F_{2,118}$ = 6.33, $P < 0.001$; *Supplementary file 2, b*). While one might generally assume that mothers behave in response to offspring solicitation behaviour, these results show, perhaps surprisingly, that variation in maternal behaviour influences the level of solicitation: here, we need to remember that there is no variation in offspring genotype so we assume that across families differences in offspring behaviour are due to differences in the genotype of their adoptive mothers.

## Coadaptation of parental and offspring traits

While in the previous section we have focused on analysing traits within foster families, we now turn to the correlation between traits of biological families, i.e. mothers and their biological offspring. This correlation has been analysed in coadaptation models, which make specific predictions about how parental and offspring traits are correlated, and in empirical work (*Kölliker et al., 2000*; *Agrawal et al., 2001*; *Curley et al., 2004*; *Lock et al., 2004*; *Kölliker et al., 2005*; *Hinde et al., 2010*; *Meunier and Kölliker, 2012*). Prior experimental studies have found both a positive (*Parus major*; *Kölliker et al., 2000*: *Nicrophorus vespilloides*; *Lock et al., 2004*) and negative correlation between offspring solicitation and parental traits (*Sehirus cinctus*; *Agrawal et al., 2001*). In our study, we found a negative correlation. When we measured short-term provisioning, we found a negative correlation between BXD offspring short-term weight gain and the corresponding

provisioning of their biological (BXD) mothers on day 10, as well as a negative correlation between BXD offspring solicitation and the corresponding provisioning of their biological (BXD) mothers on day 14 (GLM, $F_{1,32}$ = 4.77, $P$ = 0.036; $r$ = -0.34 and GLM, $F_{1,28}$ = 8.046, $P$ = 0.008, $r$ = -0.48; *Figure 5* and *Supplementary file 2, d*). Thus, our results suggest that mothers who are generous providers produce young that solicit less maternal resources than offspring born to less generous mothers. Such a negative correlation is predicted to occur when maternal traits are predominantly under selection as long as parents respond to offspring demand (which we have shown above; *Kölliker et al., 2005*). One scenario to explain this negative correlation might be that each BXD line, i.e. genotype, is characterized by a unique (to this line, everything else being equal) combination of offspring and maternal behaviours where higher maternal provisioning is correlated with lower offspring solicitation. This may be due to the cost of increased solicitation (reflected in reduced body-weight for the effort expended) for which we found evidence in our study. Bodyweight is indeed negatively correlated with the level of offspring solicitation (GLM, $F_{1,66}$ = 20.57, $P$ < 0.001 e.g. day 10, $r$ = -0.39, and day 14, $r$ = -0.44; *Figure 6* and *Supplementary file 2, e*).

## Discussion

Our study of the genetics underlying family interactions has revealed that genes expressed in offspring can indirectly influence the quality of maternal behaviour and thus offspring fitness. At the same time, we detected specific loci in maternal genotype that indirectly modify offspring traits, which shows that IGEs can be an important component of the genetic architecture of complex traits (*Bijma and Wade, 2008*). We note that while postnatal cross-fostering controls for postnatal maternal effects, prenatal maternal effects can only be addressed (to some degree) by embryo transfer (e.g. *Cowley et al., 1989*), a procedure that is impractical in genetics experiments. Potentially, this may strengthen or weaken, for example, effects of BXD genotype on B6 maternal phenotype. At the same time, pre-natal maternal effects may also contribute to a phenotypic correlation between parent and biological offspring traits, as reported above on coadaptation. In-utero effect variation due

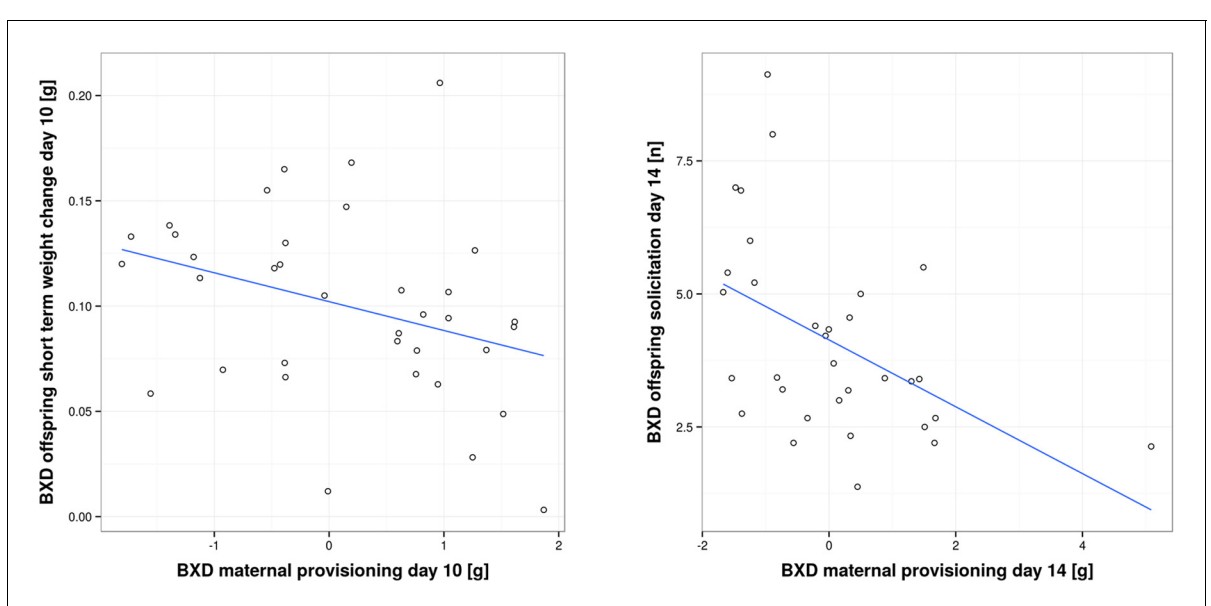

**Figure 5.** Correlation between offspring and maternal traits in biological BXD families. The first panel shows the correlation between BXD offspring short-term weight change per pup and provisioning of their corresponding biological BXD mother on day 10 per pup. The second panel shows the correlation between the level of BXD offspring solicitation per pup on day 14 and their mother's provisioning per pup.

The following source data is available for figure 5:

**Source data 1.** Source data for *Figure 5* as uploaded to GeneNetwork, showing BXD offspring short-term weight change on day 10 per pup, BXD maternal provisioning on day 10, BXD offspring solicitation on day 14 per pup and BXD maternal provisioning on day 14 for the BXD lines.

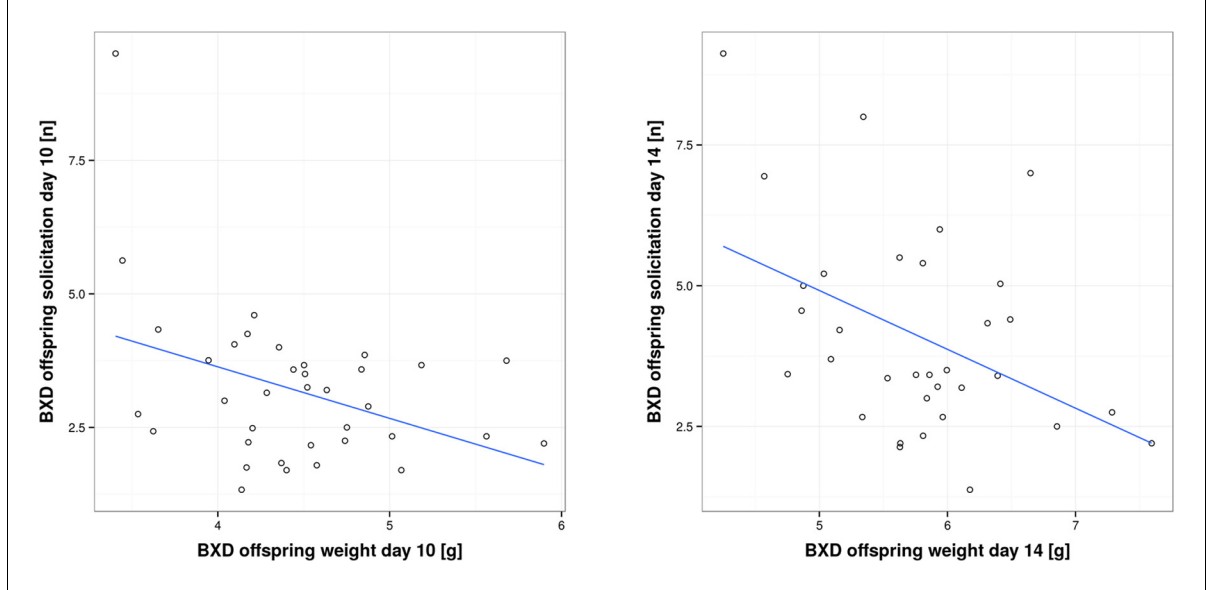

**Figure 6.** Correlation between per pup offspring solicitation and corresponding body weight in BXD lines on day 10 and day 14, respectively.

The following source data is available for figure 6:

**Source data 1.** Source data for *Figure 5* as uploaded to GeneNetwork, showing BXD offspring solicitation on day 10 per pup, BXD offspring weight on day 10 per pup, BXD offspring solicitation on day 14 per pup and BXD offspring weight on day 14 per pup for the BXD lines.

to differences in BXD genotype may therefore contribute to differences in BXD offspring behaviour, in turn affecting the behaviour of their adoptive mothers. We now need to investigate the candidates identified here and how their effects on parental and offspring traits are integrated into the gene networks determining individual development. By controlling for genetic variation in either mothers or offspring we have been able to show that levels of maternal provisioning and offspring solicitation are unique to specific genotypes (here each BXD line) and that solicitation is costly. The ability to conduct complex systems genetics analyses in experimental systems of parent offspring interactions will enable us to concentrate now on understanding the underlying pathways involved, and how they are modified by social environmental conditions that determine adult phenotypes and associated reproductive success.

## Materials and methods

### Animals and genetic analysis

We used mice of the BXD recombinant inbred population, which consists of experimentally tractable and genetically defined mouse lines capturing a large amount of naturally occurring genetic variation, which underlies variation at the phenotypic level (e.g. *Chesler et al., 2005*; *Hayes et al., 2014*). The BXD panel incorporates ~5 million segregating SNPs, 500,000 insertions and deletions, and 55,000 copy-number variants. These lines are used for complex systems genetics analyses integrating massive phenotype and gene expression data sets obtained across years and studies (e.g. *Andreux et al., 2012*; *Ashbrook et al., 2014*).

The 42 BXD strains used in this study (1, 11, 12, 14, 24, 32, 34, 38–40, 43–45, 48a, 49–51, 55, 56, 60–64, 66–71, 73a, 73b, 73–75, 83, 84, 87, 89, 90, 98, 102) were obtained from Professor Robert W. Williams at the University of Tennessee Health Science Centre, Memphis, TN. C57BL/6J (B6) mice were obtained from Charles River, UK. Three within-line repeats plus the corresponding 42 B6 families with three within-line repeats were set up for the experiment (*Figure 1*). Sample size was determined considering power analyses and logistical aspects. Mapping power is maximised with increasing number of lines whereas within-line repeats *n* increase confidence of line average

phenotypes, which, however, rapidly diminishes as *n* exceeds four (*Belknap, 1998*). We have modelled power and effect sizes following (*Belknap, 1998*): n = $(Z_\alpha + Z_\beta)2$ / $S^2QTL$ / $S^2Res$. $Z_\alpha$ and $Z_\beta$ are Z values for a given α and β; $S^2QTL$ is the phenotypic variance due to a QTL and $S^2Res$ is the residual variance. With power (1-β) of 80%, α of 0.05 we estimated that with 45 lines we can detect QTL at genome-wide significance explaining ∼ 16% of trait variance, which is sufficient mapping power given effect sizes of prior work. We have modelled the relationship between power and number of replicates using qtlDesign (*Sen et al., 2007*). Everything else being equal power can be optimized by maximizing the number of genotypes (i.e. lines) and reducing replicates, even with varying degrees of heritability. Thus, we set up three replicates using 45 lines, and line averages are mapped, although for some traits and some lines this number may be lower due to lower breeding success. Outliers have been retained as they represent distinct genotypes, evinced by outliers for different traits being from the same line. Interval mapping (*Haley and Knott, 1992*) relies on 3795 informative SNP markers across all chromosomes, except Y, as implemented in GeneNetwork (GN) (*Hager et al., 2012*). The BXD strains were genotyped using the MUGA array in 2011, along with genotypes generated earlier using Affymetrix and Illumina platforms (*Shifman et al., 2006*), and mm9 is used. Loci are identified in GN by the computation of a likelihood statistic score and significance was determined using 2000 permutations of the phenotype data. Candidates were identified within the region defined by using GeneNetwork (http://www.genenetwork.org) and further information combined from QTLminer (*Alberts and Schughart, 2010*), Entrez genes (http://www.ncbi.nlm.nih.gov/gene) and Mouse Genome Informatics (*Eppig et al., 2015*).

## Behavioural protocols

Mice were maintained under standard laboratory conditions in the same room, exclusively used for the experiment in individually ventilated cages (IVC Tecniplast Green line), and given chow and water *ad lib.* Humidity ranges between 50% and 65% relative humidity, temperature between 20°C and 21°C. All animals were kept on a reverse dark light cycle with 12h red light (active phase) and 12h white light.

Cross-fostering of entire litters took place within 24h of birth of corresponding B6 and BXD females and analyses used trait values per pup to adjust for differences in adoptive litter size, which is a significant covariate for provisioning and solicitation as of course a mother nursing a large litter will overall provide more than a mother nursing a small litter. Both mothers and litters were weighed at birth and once weekly, for three weeks until weaning, i.e. at the end of week 1, week 2 and week 3, respectively, to enable the calculation of growth during these periods. In addition, we recorded maternal and offspring behaviour on postnatal days 6, 10 and 14 when we simulated maternal departure to standardize observation conditions (*Hager and Johnstone, 2003*, *2005*). After a 4h separation, mother and litter were re-joined and maternal and offspring behaviours recorded simultaneously over 15 min, using scan sampling every 20 s (*Martin and Bateson, 2007*). Provisioning is measured using an established protocol (*Hager and Johnstone, 2003*, *2005*, *2007*) as maternal and offspring weight change after reunification with pups over the following two hours. Because rodents are nocturnal all observations occurred under red light, i.e. the active phase. Maternal behaviour was recorded as the sum of nursing, suckling and nest building. Nursing is defined as attending the litter, sitting on the nest and suckling up to half the litter while suckling refers to the entire litter being suckled at the same time. This distinction was used as sometimes it cannot be ascertained whether pups are suckling or not because of the position of the mothers in the nest. Nestbuilding behaviour is gathering nesting material and constructing a nest. Pup solicitation behaviour in mice is defined as pups attempting to suck and following the mother, but individual pups were not distinguished. All procedures were approved by the University of Manchester Ethics Committee.

## Acknowledgements

We would like to thank Tucker Gilman, John Fitzpatrick, Casey Bergman and three anonymous reviewers for their thoughtful comments on a previous draft of the manuscript. DGA and BG are supported by the Biotechnology and Biological Sciences Research Council (BBSRC), UK and NERC grant NE/F013418/1.

## Additional information

### Funding

| Funder | Grant reference number | Author |
| --- | --- | --- |
| Natural Environment Research Council | NE/F013418/1 | Reinmar Hager |
| Biotechnology and Biological Sciences Research Council | | David George Ashbrook |

The funders had no role in study design, data collection and interpretation, or the decision to submit the work for publication.

### Author contributions

DGA, Analysis and interpretation of data, Drafting or revising the article; BG, Conception and design, Acquisition of data, Analysis and interpretation of data; RH, Conception and design, Acquisition of data, Analysis and interpretation of data, Drafting or revising the article

### Author ORCIDs

David George Ashbrook, http://orcid.org/0000-0002-7397-8910

### Ethics

Animal experimentation: All procedures were approved by the University of Manchester Ethics Committee.

## Additional files

### Supplementary files

• Supplementary file 1: A) Functional and further details about the genes within the *Osplge5.1* QTL for B6 maternal behaviour on day 14, obtained from GeneNetwork, Entrez genes, and Mouse Genome Informatics. Potential candidate genes in this region are *Cyp3a16, Cyp3a44, Cyp3a11, Cyp3a25* and *Cyp3a41a* because of their involvement in steroid hormone biosynthesis. B) Functional and further details about the genes within the *MatDge1.1* QTL for BXD nestbuilding on day 6, obtained from GeneNetwork, Entrez genes, and Mouse Genome Informatics. Potential candidate genes in this region include: *Hsd17b7*, as it is involved in steroid hormone synthesis, and hormonal regulation is needed to initiate nestbuilding behaviour (*Gammie et al., 2007*; *Keisala et al., 2007*; *Bester-Meredith and Marler, 2012*); *Pou2f1, Lmx1a* and *Rgs4* as they are linked to activity related phenotypes; finally, *Pbx1* and *Ddr2* have been linked to craniofacial morphology, which may affect the ability to make nests (*Schneider et al., 2012*). C) Functional and further details about the genes within the *MatDge10.1* QTL for BXD maternal behaviour on day 6, obtained from GeneNetwork, Entrez genes, and Mouse Genome Informatics. A potential candidate gene is *Ifngr1* as it is related to depression-like behaviour, which in turn has been linked to reduced maternal care (*Smith et al., 2004*; *Malkesman et al., 2008*). D) Functional and further details about the genes within the *OspDge5.1* QTL for BXD offspring solicitation on day 6, obtained from GeneNetwork, Entrez genes, and Mouse Genome Informatics. *Cdk5* might be a good candidate as it is involved in several neuronal annotations (e.g. axonogenesis and synaptic transmission), and mutants have no suckling reflex (*Ohshima et al., 1996*).

• Supplementary file 2: Details of correlation analyses.

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
