## [Decision Letter]

Thank you for submitting your work entitled "Genetic variation in offspring indirectly influences the quality of maternal behaviour in mice" for consideration by eLife. Your article has been favorably evaluated by Diethard Tautz (Senior editor) and three reviewers, one of whom is a member of our Board of Reviewing Editors.

The reviewers have discussed the reviews with one another and the Reviewing editor has drafted this decision to help you prepare a revised submission.

Summary:

This paper demonstrates that indirect genetic effects can be identified and genetically mapped. The authors adopt a cross-fostering design so that in families either the mothers or the offspring are genetically variable (from the BXD recombinant inbred panel) while the corresponding offspring or mothers show no genetic variation (i.e. are from the B6 inbred strain). The authors identify three indirect genetic effects, and three direct genetic effects. Overall this study represents a substantial and well-designed effort to address an important question.

Essential revisions:

1) Design limitations need to be acknowledged and discussed. In their design the in-utero maternal effects are perfectly confounded with offspring genotypes. This means that the results of part 1 could be interpreted as "Variation in BXD mothers genotypes causes variation in in-utero environment, which causes variation in BXD offspring behaviour later in early life, which causes variation in B6 adoptive mothers provisioning". This would effectively be IGE from BXD mothers on B6 mothers.

The results of part 2 could be interpreted as "Variation in BXD mothers genotypes causes variation in in-utero environment, which causes variation in BxD offspring weight gain and solicitation in early life" and Variation in BXD mothers genotypes independently causes variation in BXD biological mothers provisioning, in which case there would not be coadaptation between BXD offspring phenotypes and BXD maternal behaviour.

This caveat is sufficiently important that its implications should be discussed in the Discussion. As the authors note in the Discussion however, overcoming this limitation would be extremely challenging.

2) The results are only just significant. Given that they have tested multiple behaviors, and both direct and indirect effects, the chances are that a proportion of the loci are likely false positives. The authors should assess a false discovery rate and provide results corrected for the total number of tests carried out.

3) The mapping approach is relatively un-sophisticated and in particular environmental effects receive scant attention. Cage effects should be accounted for in all the analyses. In addition there may be unacknowledged environmental effects, which might be captured if they included the time when the experiment was carried out as a covariate. The fact that results for the genetic effects only just achieve significance means that there is a concern that even slight contributions from covariates may be giving rise to false positives.

4) It wasn't clear why they used per pup values. Why not fit litter size should be fitted as a covariate? They should repeat the analyses with litter fitted as a covariate.

5) The phenotypic correlation observed between BXD offspring solicitation and B6 maternal behaviour could arise from environmental effects only as BXD offspring and B6 mothers share a cage. It is not strong evidence that BXD genotypes affect B6 maternal behaviour, and can only weakly be linked to the QTLs identified as no overlap was found between DGE QTLs for offspring solicitation and IGE QTLs for maternal behaviour. Genetic correlation should be computed from the individual measurements.

[Editors' note: further revisions were requested prior to acceptance, as described below.]

Thank you for resubmitting your work entitled "Genetic variation in offspring indirectly influences the quality of maternal behaviour in mice" for further consideration at *eLife*. Your revised article has been favorably evaluated by Diethard Tautz (Senior editor) and three reviewers, one of whom is a member of our Board of Reviewing Editors. The manuscript has been improved but there are some remaining issues that need to be addressed before acceptance, as outlined below:

You state that you have "used the correct litter size to calculate per pup measures, namely adoptive litter size." That a litter size effect exists does not imply that per pup values are appropriate, i.e., it does not prove that per pup values are independent of litter size. This would only be correct if you could prove that total provisioning is exactly linear in litter size; the mere significance of a (linear) litter size effect is no proof thereof.

Intuitively, one would expect that provisioning per pup decreases with litter size, simply because there is some sort of limit to total provisioning that is feasible. If you can fit a litter size effect on total provisioning using linear models, then you can also do this on provisioning defined per pup.

While this may not be feasible in the Genenetwork software that you used for the mapping, it could be investigated in a prior analysis using simple linear models. Then the results should be reported in the manuscript, and if the litter size effect on per pup provisioning is significant in the prior analysis, then you should acknowledge in the manuscript that ideally a covariate for litter size would have been included in the mapping.

The correlation/covariance that may exist between direct and indirect environmental effects (IDE and IEE) should be modelled when calculating the correlation between mother and offspring phenotypes as it may increase or decrease the correlation reported in the manuscript.

Example of such a correlation between IDE and IEE: differences in noise across cages may lead to B6 mothers in noisy cages showing reduced maternal behaviour and BXD offspring in those same cages soliciting less (or more).

To account for this covariance, you would need to focus on the genetic component of the correlation and use the replicate structure of the data (3 replicates per genotype) to calculate something analogous to a genetic correlation but where phenotypes are measured in B6 mothers and BXD offspring (instead of in the same individual as is traditional) and the genotypes would be those of the BXD offspring.

Although a sentence was added in the Discussion with regard to in-utero effects, you still need to put the results of part 1 in perspective and address interpretation of the results of part 2 (co-adaptation).

---

## [Author Response]

*Essential revisions:*

*1) Design limitations need to be acknowledged and discussed. In their design the in-utero maternal effects are perfectly confounded with offspring genotypes. This means that the results of part 1 could be interpreted as "Variation in BXD mothers genotypes causes variation in in-utero environment, which causes variation in BXD offspring behaviour later in early life, which causes variation in B6 adoptive mothers provisioning". This would effectively be IGE from BXD mothers on B6 mothers.*

*The results of part 2 could be interpreted as "Variation in BXD mothers genotypes causes variation in in-utero environment, which causes variation in BxD offspring weight gain and solicitation in early life" and Variation in BXD mothers genotypes independently causes variation in BXD biological mothers provisioning, in which case there would not be coadaptation between BXD offspring phenotypes and BXD maternal behaviour.*

*This caveat is sufficiently important that its implications should be discussed in the Discussion. As the authors note in the Discussion however, overcoming this limitation would be extremely challenging.*

We agree, and have thus expanded the first paragraph of the Discussion following this suggestion though we note coadaptation may well be manifest in utero (see e.g. Wolf and Hager 2006 PLoS Biol 10.1371/journal.pbio.0040380, and 2009 in J Evol Biol 22:2519).

*2) The results are only just significant. Given that they have tested multiple behaviors, and both direct and indirect effects, the chances are that a proportion of the loci are likely false positives. The authors should assess a false discovery rate and provide results corrected for the total number of tests carried out.*

The experiment was designed to test the hypothesis that genetic variation in BXD offspring can affect trait variation in B6 maternal behaviour and vice versa in the context of family interactions. While we need protection against many false discoveries we need, at the same time, assurance we are not neglecting truly interesting locations (see e.g. Chen and Storey 2005 Genetics 173: 2371–2381). Traditionally, LOD scores above 3.3 are considered in interval mapping studies and we follow the convention of previous studies on maternal behaviour (e.g. Peripato et al. 2002 Genetics 162:1341) and other multiple complex traits (Cheverud et al. 2008 PNAS 105:4253, Wolf et al. 2008 PLoS Genet 4:e1000091, Hager et al. 2012 Nat Commun 3:1079) in reporting significance both at the genome-wide and suggestive level, which we exceed with five offspring and maternal traits for all loci (0.63/5 or 10,respectively).

In addition, our result that variation in maternal behaviour is affected by our identified (IGE) in offspring is further supported by the phenotypic correlation: “offspring solicitation behaviour in […] BXD pups is positively correlated with the level of maternal behaviour […] in B6 adoptive mothers on all three days we measured behaviour with significant effects of maternal behaviour and day on offspring solicitation[…]” (Results, second paragraph), notwithstanding the caveat added in the Discussion about in-utero effects.

*3) The mapping approach is relatively un-sophisticated and in particular environmental effects receive scant attention. Cage effects should be accounted for in all the analyses. In addition there may be unacknowledged environmental effects, which might be captured if they included the time when the experiment was carried out as a covariate. The fact that results for the genetic effects only just achieve significance means that there is a concern that even slight contributions from covariates may be giving rise to false positives.*

Please see next point.

*4) It wasn't clear why they used per pup values. Why not fit litter size should be fitted as a covariate? They should repeat the analyses with litter fitted as a covariate.*

In reply to points 3 and 4: First, we need to clarify the per pup measure used as it appears that this will answer several points raised. We did not record individual pup behaviour (where cage effects may appear) but rather for the entire litter and the per pup measure divides the score by litter size. This is because mothers may differ e.g. in their provisioning not because they have different genotypes but simply because they nurse litters of different sizes. Using linear models we found significant litter size effects. Thus, using a per pup measure is fully justified and appropriate, as well as supported by statistics, and we have used the correct litter size to calculate per pup measures, namely adoptive litter size. We have clarified this in the Methods section now. All our analyses are conduced within Genenetwork, the analytical tool kit for the BXD population (www.genenetwork.org) used in over 1000 papers, which at present does not allow running models with covariates but we understand this will be implemented in future.

While we unfortunately do not have the time of each recording we have experimentally ensured as much as possible to standardize motivation in mothers and offspring. This is a technique we have employed and developed over years (e.g. Hager and Johnstone 2003 Nature 421:533, 2005 Ethology 111:705, 2007 Anim Behav 74:139, Lyst et al. 2012 PLoS One 7:e47640) for the following reasons: consider the scenario of measuring nursing behaviour. Prior to the experiment on the day we have no information about whether mothers have just been nursing their pups or whether the last bout of nursing was several hours ago. Thus, we might see a difference in the pups’ motivation to suck, and in the mothers to nurse, not because of differences in genotypes but because of differences in the times that elapsed between the last feed. Therefore, we standardize the time prior to us measuring nursing.

*5) The phenotypic correlation observed between BXD offspring solicitation and B6 maternal behaviour (part 1.2) could arise from environmental effects only as BXD offspring and B6 mothers share a cage. It is not strong evidence that BXD genotypes affect B6 maternal behaviour, and can only weakly be linked to the QTLs identified as no overlap was found between DGE QTLs for offspring solicitation and IGE QTLs for maternal behaviour. Genetic correlation should be computed from the individual measurements.*

Cage effects may arise by a shared environment among multiple offspring that have been individually phenotyped but this is not how offspring traits were measured (see previous comment). Given that one family (i.e. one BXD genotype plus B6) occupied one cage we do not think that this effect applies here, and we have published correlations using the same protocol multiply before (e.g. Hager and Johnstone 2003 Nature 421:533, 2005 Ethology 111:705, 2006a,b Biol Lett 2:81 and 2:253). Further, our result that variation in maternal behaviour is affected by genes expressed in offspring is supported by phenotypic results: offspring solicitation behaviour in genetically variable BXD pups is positively correlated with the level of maternal behaviour in their genetically uniform B6 adoptive mothers on all three days we measured behaviour with significant effects of maternal behaviour and day on offspring solicitation. We appreciate the point the reviewer is making regarding an overlap of DGE QTL and IGE QTL, but no direct support is provided and we have no reason to assume that DGE QTL and IGE QTL have to overlap to support this argument.

[Editors' note: further revisions were requested prior to acceptance, as described below.]

*Intuitively, one would expect that provisioning per pup decreases with litter size, simply because there is some sort of limit to total provisioning that is feasible. If you can fit a litter size effect on total provisioning using linear models, then you can also do this on provisioning defined per pup. While this may not be feasible in the Genenetwork software that you used for the mapping, it could be investigated in a prior analysis using simple linear models. Then the results should be reported in the manuscript, and if the litter size effect on per pup provisioning is significant in the prior analysis, then you should acknowledge in the manuscript that ideally a covariate for litter size would have been included in the mapping.* We have run linear models as suggested and can confirm that litter size is a non-significant predictor for all our per pup provisioning traits.

*The correlation/covariance that may exist between direct and indirect environmental effects (IDE and IEE) should be modelled when calculating the correlation between mother and offspring phenotypes as it may increase or decrease the correlation reported in the manuscript.*

*Example of such a correlation between IDE and IEE: differences in noise across cages may lead to B6 mothers in noisy cages showing reduced maternal behaviour and BXD offspring in those same cages soliciting less (or more). To account for this covariance, you would need to focus on the genetic component of the correlation and use the replicate structure of the data (3 replicates per genotype) to calculate something analogous to a genetic correlation but where phenotypes are measured in B6 mothers and BXD offspring (instead of in the same individual as is traditional) and the genotypes would be those of the BXD offspring.*

The example given that a correlation between B6 maternal behaviour and BXD offspring solicitation could arise because all B6 shared the same environment (e.g. noise) is a very unlikely scenario as all animals are kept in the same small room (exclusively used by us), which is sound proofed, and in IVC cages, each with the same conditions. We note that the conditions experienced by B6 females are identical except the genotypes of the adoptive BXD litters, and that correlations are calculated per maternal genotype, not cage; and have added some additional information about housing to the Methods.

*Although a sentence was added in the Discussion with regard to in-utero effects, you still need to put the results of part 1 in perspective and address interpretation of the results of part 2 (co-adaptation).*

Yes, agreed. We have added potential implications of in-utero effects to that section in the Discussion.